# NEURAL NETWORK PARAMETER REGRESSION FOR LATTICE QUANTUM CHROMODYNAMICS SIMULATIONS IN NUCLEAR AND PARTICLE PHYSICS

**Phiala E. Shanahan**
Department of Physics
William & Mary
Williamsburg, VA 23187-8795, USA
Jefferson Laboratory
Newport News, VA 23606, USA
peshanahan@wm.edu

**Daniel Trewartha**
Jefferson Laboratory
Newport News, VA 23606, USA
danielt@jlab.org

**William Detmold**
Center for Theoretical Physics
Massachusetts Institute of Technology
Cambridge, MA 02139, USA
wdetmold@mit.edu

## ABSTRACT

Nuclear and particle physicists seek to understand the structure of matter at the smallest scales through numerical simulations of lattice Quantum Chromodynamics (LQCD) performed on the largest supercomputers available. Multi-scale techniques have the potential to dramatically reduce the computational cost of such simulations, if a challenging parameter regression problem matching physics at different resolution scales can be solved. Simple neural networks applied to this task fail because of the dramatic inverted data hierarchy that this problem displays, with orders of magnitude fewer samples typically available than degrees of freedom per sample. Symmetry-aware networks that respect the complicated invariances of the underlying physics, however, provide an efficient and practical solution. Further efforts to incorporate invariances and constraints that are typical of physics problems into neural networks and other machine learning algorithms have potential to dramatically impact studies of systems in nuclear, particle, condensed matter, and statistical physics.

## 1 DATA

LQCD is a Markov Chain Monte-Carlo (MCMC) importance sampling method based on the generation of ensembles of configurations of the underlying physical degrees of freedom (quark and gluon fields) Rothe (1992). Configurations are represented as sets of links $U_\mu(x)$ between sites on four-dimensional hypercubic space-time lattices ($x$ denotes the spacetime coordinates of the origin site and $\mu$ the direction of the link). Each link can be encoded as an SU(3) matrix (a $3 \times 3$ complex matrix $M$ with $M^{-1} = M^\dagger$ and $\det[M] = 1$, where $M^\dagger = (M^*)^T$ is the Hermitian conjugate), and a configuration is encoded by $\mathcal{O}(10^7)$ links, i.e., $\mathcal{O}(10^9)$ floating point or double precision numbers for a typical state-of-the-art calculation. Since these configurations sample the probability distribution corresponding to the LQCD action (a function defining the quark and gluon dynamics), weighted ensemble averages determine physical observables of interest; calculations typically use ensembles of $\mathcal{O}(10^3)$ configurations.

The parameter regression task studied here is the determination of the LQCD action parameters, $\beta$ and $m$, corresponding to a given ensemble of configurations. Because of the significantly inverted data hierarchy of LQCD datasets, this is a challenging problem. However, the physics encoded by an ensemble of configurations is invariant under a number of complex symmetries, namely discrete space-time translations and rotations on the hypercubic space, as well as 'gauge transformations'. The latter are continuous Lie group transformations at each space-time point on the lattice, i.e.,

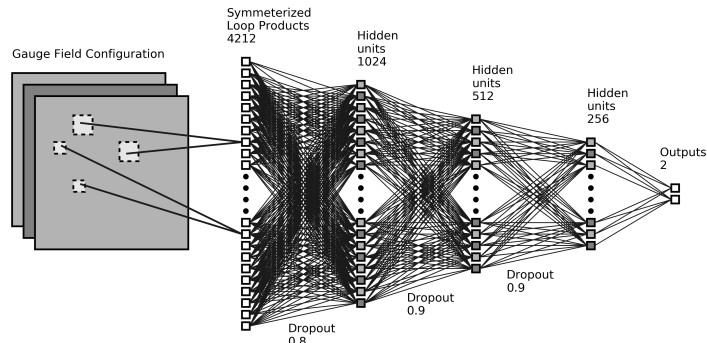

Figure 1: Neural network structure. In the first layer, a total of 4212 symmetry invariant products are formed. Fully connected hidden layers with 1024, 512, and 256 nodes act on these features. Each hidden layer uses a `tanh` activation function, with dropouts between layers.

$U_\mu(x) \to U'_\mu(x) = \Omega(x)U_\mu(x)\Omega^\dagger(x+\hat{\mu})$, where $\Omega(x)$ represents a set of arbitrary SU(3) matrices, one for each space-time location $x$, and $\hat{\mu}$ denotes a unit displacement in the $\mu$ direction. Some of these symmetries only manifest stochastically after ensemble averaging. Encoding the relevant invariances within network structures, or equivalently using bases of symmetry-invariant input features Simard et al. (1992); Király et al. (2014); Schölkopf et al. (1996); Burges & Schölkopf (1997), reduces the degrees of freedom of the problem and mitigates the severity of the data hierarchy. Using such an approach allows successful and efficient regression of action parameters from LQCD datasets (other approaches such as data augmentation to accelerate the stochastic learning of these symmetries are computationally infeasible).

## 2 RESULTS

Since LQCD data is generated using MCMC, it is computationally cheaper to produce large ensembles at a few sets of action parameters rather than single samples at many parameter values well-distributed in the space of interest. For the regression exercise, training (validation) datasets were constructed by randomly selecting 850 (100) configurations from each of twenty ensembles generated with LQCD action parameters $\beta$ and $m$ in a regular grid. Trained on this data set, a variety of fully-connected network structures produced precise and accurate predictions of the action parameters from the validation ensembles. However, these models failed to generalise to intermediate parameter values, always returning the mean of the training parameter grid. This indicates that the models are overfitting the data, a result that might be expected given the challenging data hierarchy. Although ultimately unsuccessful at the task at hand, these models identified a previously unknown feature of the LQCD configurations with a longer MCMC autocorrelation time than any known quantity.

In order to overcome the data hierarchy problem, neural networks trained using symmetry-preserving features of reduced dimension were also considered. The structures that were constructed, called Wilson loops, correspond to closed paths of links ($U_\mu(x)$) of various shapes and sizes. Optimally, the selection of a set of such features would be part of the training prescription. Because of the significant computational cost associated with the calculation of each possible symmetry-preserving feature on each configuration, however, several categories of Wilson loops were chosen by hand for this study. Network structures such as that illustrated in Fig. 1 were trained on the Wilson loop feature sets constructed on the training data described above. Although no rigorous hyperparameter tuning was undertaken, many variations of this network structure, including various numbers of nodes and layers, different activation functions (`tanh`, `reLU`, and `sigmoid`), and different choices of dropout and normalisation hyperparameters were used. The best-performing network configuration is shown and its output is detailed here. Various minimisation algorithms, including stochastic gradient descent, Adam Kingma & Ba (2014), and Nesterov Nesterov (1983), with various parameters, achieved the same minimum loss, although the numbers of epochs to achieve convergence varied. An L1 loss function in the two-dimensional parameter space outperformed L2 for regression performance in all tests. Network biases were initialised to zero, and

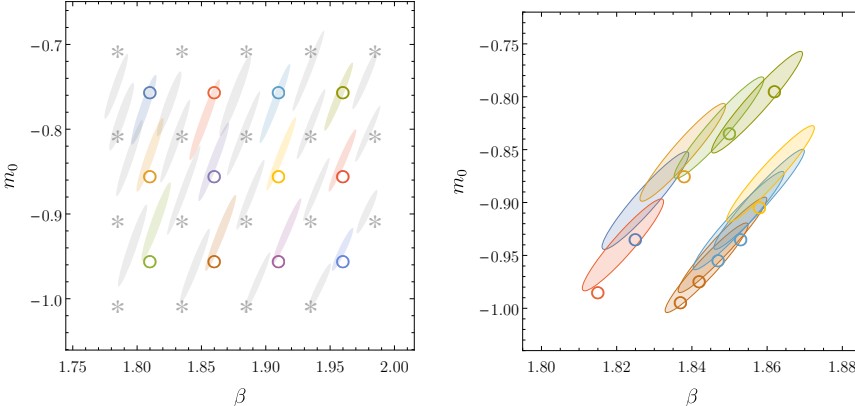

Figure 2: Parameter predictions for intermediate validation ensembles. The open circles show parameters of each validation ensemble, while the ellipses show the one–standard-deviation confidence regions generated using the parameter predictions for the 100 validation configurations from each ensemble. The greyed-out stars and ellipses show the training ensemble parameters and the confidence regions for validation ensembles at the parameters of the training data.

weights to a truncated normal distribution centred at zero with a width of 0.02. An L2 regulariser with weight decay 0.001 was used.

The predictions generated by a trained network instance are shown in Fig. 2. As well as achieving accurate and precise parameter predictions for the validation datasets at the same parameters as the training data, the model generalises successfully to intermediate action parameter values. The elongation of the prediction ellipses is in the direction of constant $1 \times 1$ Wilson loop, the simplest and most precise gauge-invariant object that can be formed. It should be noted that there is a maximum possible regression performance for this set-up: individual gauge configurations are sampled from a probability distribution and any given configuration could, with some probability, have been generated from a range of actions with different parameters. This maximum resolution would be sharpened by batched or ensemble-based regression, which will be investigated in future work. A particularly challenging test of network performance is provided by two sets of validation ensembles which were constructed to have constant physics properties, separated in parameter space at distances much smaller than the training grid spacing. Under a principal component analysis, configurations from ensembles in either of the sets cannot be distinguished. As illustrated in Fig. 2, however, the parameter predictions from the trained network for the different datasets are distinguishable, and the central values have the correct relative positions in parameter space. This is a definitive success, indicating that the network has accurately parameterised the relevant features of the LQCD data.

## 3 SUMMARY

Neural networks trained on symmetry-invariant features solve a challenging parameter regression task in LQCD, overcoming the dramatic inverted data hierarchy natural to such problems which have orders of magnitude fewer samples available than the number of real numbers describing each sample. Further study of efficient implementations of domain knowledge by imposing complicated invariances and constraints into network structure, rather than feature selection, will have applications not only to these problems, but to related studies in statistical and condensed matter physics.

ACKNOWLEDGMENTS

We are grateful to Kyle Cranmer, Michael Endres, Brendan Fong, Andrew Pochinsky, and Mike Williams for numerous discussions. The calculations in this project were performed using the Hyak High Performance Computing and Data Ecosystem at the University of Washington (NSF Grant Number 0922770). This work was partly supported by U.S. Department of Energy (DE-SC0010495, DE-SC0011090, DE-SC0018121, DE-AC0506OR23177) and by the Exascale Computing Project (17-SC-20-SC).

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
