# OpenReview forum: "Neural network parameter regression for lattice quantum chromodynamics simulations in nuclear and particle physics"
_ICLR.cc/2018/Workshop — Accept_

### Official Review · AnonReviewer3 · 2018-03-09
**examines invariance-aware structure for one real-world problem, but generality and significance vs baselines not clearly established**

**Rating:** 6
**Confidence:** 4

**Review:**

Despite claims in the abstract that such techniques might “dramatically reduce the computational cost”, the experiments actually offered are not convincing, and not examined sufficiently for the audience to appreciate the lessons learned and to be extracted from such an applied paper.
Their main contribution seems to be the use of “Wilson loop” structures to capture invariances in their task.  It is not obvious that this approach will be particularly useful or insightful for any applications different than the specific / esoteric one of interest to the authors.  They claim that a generic fully connected network without such special structure “failed to generalize .. always returning the mean of the training parameter grid”.  To convince the reader that their attempt to use generic FC DNNs was fair, the “variety” of network structures and optimization approaches they tried should be concisely reported. They said they suspected that the nets overfit, which begs the question of whether higher bias models (even simple linear models) might have provided better baselines to evaluate their proposed approach, than high capacity FC DNNS.  Also, Section 2 mentions examining a variety of hyper parameters, but doesn’t make it clear whether both generic and Wilson loop based nets were given equal opportunities to exploit such hyper parameter search. In short, the authors seem to have found an effective solution to capture the invariances of their specific task, but have not provided very convincing evidence that their complex/unusual approach was necessary, nor whether this approach might generalize to other problems beyond their very specific one.  It is not clear what the ICLR audience should take away from such a paper.

Given this is only a workshop, a more lenient rating than for a conference paper is given. But even then, the paper (and potential to spark meaningful discussions at a workshop) would benefit greatly from addressing the above concerns.

---

### Official Review · AnonReviewer1 · 2018-03-10
**Thoughtful application of neural networks to a consequential problem in physics**

**Rating:** 7
**Confidence:** 3

**Review:**

Creating neural networks with useful invariance properties is a critically important area of research.  The authors propose to use symmetry networks for a compelling application, with interesting preliminary results.  This short paper also contains a thoughtful discussion of desired network properties for this application.  This is nice early work.

---

### Official Review · AnonReviewer2 · 2018-03-11
**cool physics-based data augmentation; promise to significantly speed up simulations**

**Rating:** 7
**Confidence:** 3

**Review:**

this is a nice result: by augmenting simulated lattice quantum chromodynamics simulation data, using prior knowledge about symmetries in the configurations, action parameters can be accurately inferred on test data. this reduces the computational burden and may allow us to explore novel physics in the future. the technique consists of sampling configurations from markov chain monte carlo, then augmenting this simulation-generated dataset using some categories of wilson loops (closed paths of links). building in this prior knowledge reduces the amount of data needed to train neural network models to regress the action parameters.

points of confusion:

please clarify the problem setup. it's accessible to a physicist but not a machine learner.

1) specify what the data is. it's confusing to the average machine learner that you generate the data using MCMC.

2) how is this not data augmentation? ML people are used to this concept, so if you don't like the name for taxonomical reasons, maybe call it feature augmentation (i see that you contrast it with data augmentation; but stretching/skewing of images is referred to as data augmentation; these transformations act on features (rgb values in jpeg files) and not on 'raw' or lossless image formats)?

* consider adding an algorithm box for how you augment the data to preserve the symmetries/invariances? this will make your technique clearer

* what is beta, m? assume beta is inverse temperature? this will help make the problem concrete

---

### Decision · Program_Chairs · 2018-03-20
**ICLR 2018 Workshop Acceptance Decision**

**Decision:**

Accept

**Comment:**

Congratulations, your paper was accepted to the ICLR workshop.